# Epigenetic Age Acceleration in Mothers and Offspring 4–10 Years after a Pregnancy Complicated by Gestational Diabetes and Obesity

**DOI:** 10.3390/metabo12121226

**Published:** 2022-12-07

**Authors:** Nita Kanney, Amit Patki, Paula Chandler-Laney, W. Timothy Garvey, Bertha A. Hidalgo

**Affiliations:** 1Department of Epidemiology, University of Alabama at Birmingham, Birmingham, AL 35294, USA; 2Department of Biostatistics, University of Alabama at Birmingham, Birmingham, AL 35294, USA; 3Department of Nutrition Sciences, University of Alabama at Birmingham, Birmingham, AL 35294, USA

**Keywords:** DNA methylation, Horvath clock, epigenetics, age acceleration, gestational diabetes mellitus, lipids, glucose

## Abstract

A known association exists between exposure to gestational diabetes mellitus (GDM) and epigenetic age acceleration (EAA) in GDM-exposed offspring compared to those without GDM exposure. This association has not been assessed previously in mothers with pregnancies complicated by GDM. A total of 137 mother-child dyads with an index pregnancy 4–10 years before study enrollment were included. Clinical data and whole blood samples were collected and quantified to obtain DNA methylation (DNAm) estimates using the Illumina MethylEPIC 850K array in mothers and offspring. DNAm age and age acceleration were evaluated using the Horvath and Hannum clocks. Multivariable linear regression models were performed to determine the association between EAA and leptin, high-density lipoprotein cholesterol (HDL-C), fasting glucose, fasting insulin, and HOMA-IR. Mothers with a GDM and non-GDM pregnancy had strong correlations between chronological age and DNAm age (r > 0.70). Offspring of GDM mothers had moderate to strong correlations, whereas offspring of non-GDM mothers had moderate correlations between chronological age and DNAm age. Association analyses revealed a significant association between EAA and fasting insulin in offspring (FDR < 0.05), while HDL-C was the only metabolic marker significantly associated with EAA in mothers (FDR < 0.05). Mothers in the GDM group had a higher predicted epigenetic age and age acceleration than mothers in the non-GDM group. The association between EAA with elevated fasting insulin in offspring and elevated HDL-C in mothers suggests possible biomarkers that can better elucidate the effects of exposure to a GDM pregnancy and future cardiometabolic outcomes.

## 1. Introduction

Pregnant individuals with gestational diabetes mellitus (GDM) exhibit elevated plasma glucose levels and, in some instances, a greater risk for poor pregnancy outcomes [1]. In the U.S., the estimated prevalence of GDM occurs in 6 to 10% of pregnancies [2,3]. GDM also accounts for an elevated risk of health complications during and after pregnancy [4], including type 2 diabetes (T2D) and cardiovascular disease (CVD) [3,5,6]. Multiple studies have examined the implications a GDM pregnancy has on offspring health, including higher average systolic blood pressure Z score, increased CVD risk, and an increased prevalence of hypertension among offspring born to mothers with a GDM pregnancy compared to non-GDM-exposed offspring [7,8,9]. Additionally, studies have shown significant associations between childhood obesity and adiposity with exposure to GDM in utero [10,11].

A GDM pregnancy alone is not deterministic of adverse outcomes in the mother’s and offspring’s lifetime. DNA methylation (DNAm)—chemical modifications of DNA (at cytosine-phosphate-guanine (CpG) dinucleotides) that are independent of the underlying sequence—regulate gene expression and provide additional opportunities to estimate risks associated with GDM exposure. In pregnancies complicated by GDM, one study found over 2000 significant differentially methylated CpG sites in the fetal cord blood of GDM-exposed infants [12]. Other studies have examined DNAm differences between GDM-exposed and non-GDM-exposed offspring. Awamleh et al. examined genome-wide DNAm differences between GDM-exposed and unexposed offspring [13]. From cord blood and placenta samples, they found 662 and 99 significant CpG sites that were differentially methylated between GDM-exposed and non-GDM-exposed newborns, respectively. Similarly, Finer et al. identified 1485 significant CpG sites in offspring cord blood and 1708 in the placenta as methylation variable positions [14]. GDM-exposed mothers had an additional 238 sites in cord blood and 479 sites in the placenta that were hypermethylated compared to non-GDM-exposed mothers.

To date, only two studies have investigated the relationship between DNAm age, epigenetic age acceleration (EAA), and associations of EAA with metabolic biomarkers in GDM- and non-GDM-exposed offspring [15,16]. Both studies reported a higher EAA in GDM-exposed offspring compared to unexposed. In the paper by Kim et al., accelerated extrinsic epigenetic aging (EEAA) in GDM offspring was associated with increased insulin resistance and secretion compared to unexposed offspring [15]. Shiau et al. reported that a higher age acceleration in GDM-exposed offspring was associated with increased systolic and diastolic blood pressure, hemoglobin, and Suprailiac skinfold compared to unexposed offspring [16]. A better understanding of these associations is warranted, including the opportunity to interrogate the epigenome of both mothers and offspring in a single study. Therefore, our study aims to expand these investigations by examining the association of EAA with five metabolic biomarkers of interest and GDM by calculating epigenetic age as a comparator to chronological age in mothers with and without GDM and their offspring from the Health After Pregnancy—Intergenerational Transmission of Obesity (HAPi) study.

## 2. Materials and Methods

### 2.1. Study Population

A total of 137 mother-child dyads who had an index pregnancy 4–10 years ago prior to enrollment in the HAPi study were included in this analysis. Information on participant recruitment, selection, methods used to screen for GDM, self-reported demographic information (including race/ethnicity), concomitant diseases, and medication use was previously described [17]. Women with hepatitis B or C, lupus, heart disease, HIV, or renal disease during their index pregnancy were excluded from the study, as were those with opioid, other narcotic, or illicit drug use, or tobacco or alcohol use during the index pregnancy. Initial recruitment began in April 2017 until June 2019, including mothers who had received prenatal care at the University of Alabama at Birmingham (UAB) and had given birth between 2007 and 2015. Selection into the study cohort was based on a determination of the mother’s GDM status at the time of delivery. Confirmation of whether mothers had GDM during their pregnancy was conducted through a prenatal record review for mothers who received care at UAB. Alternatively, mothers who gave birth outside of UAB provided their GDM status through self-report, which was later verified through a review of their medical records. At their baseline visit, mothers with hepatitis B or C, lupus, heart disease, HIV, and renal disease were excluded. Additionally, pregnancies complicated by maternal use of opioids or other illicit drug use, tobacco, or alcohol were not included in the study. To ensure proper selection into the unexposed group, normal weight (NW) mothers were screened for certain conditions. NW mothers who were diagnosed with glycosuria, preeclampsia, hypertension, impaired glucose tolerance, or type 2 diabetes after giving birth were excluded from participation in the study. Children were excluded from participating in the study if they were growth restricted in utero, had type 1 diabetes, or exhibited a serious health condition, including congenital heart disease. Approval for the study was granted by the Institutional Review Board for Human Use at the University of Alabama at Birmingham.

### 2.2. Clinical Measurements

Classification of mothers as NW and GDM was based on established guidelines that have been previously described [18,19]. Mothers who were previously overweight or obese (BMI ≥ 25.0 kg/m^2^) and had GDM during the index pregnancy (n = 67) were selected into the exposed or GDM group, while mothers who had a prior BMI < 25.0 kg/m^2^ and did not have a prior GDM pregnancy (n = 76) were in the NW group. However, for the purposes of our study, NW mothers were referred to as the non-GDM group. Procedures to obtain anthropometrics (waist-to-hip ratio, BMI, body fat percent) and metabolic biomarkers (Leptin, HDL-C, HOMA-IR) for mothers and offspring have been described previously [17]. Measurements were taken to provide information on the body fat percent of mothers [20]. Similar measurements were taken from children, including their BMI z-score [21] and body fat percentage [22]. Assays were performed to obtain measures of the adipokine leptin, high-density lipoprotein cholesterol (HDL-C), homeostatic model assessment of insulin resistance (HOMA-IR), fasting glucose, and fasting insulin [17]. Tanner stage classification for children was based on a brief physical exam by a trained registered nurse and established criteria for pubertal development [23,24].

### 2.3. DNA Methylation Measurements

DNA was extracted from samples of whole blood obtained from mothers and their offspring. DNA quantification and quality estimation were performed prior to DNAm quantification. Extracted DNA was assayed utilizing the Infinium MethylationEPIC BeadChip kit at the University of Minnesota Genomics Center. Quality control was carried out using Illumina’s GenomeStudio software. Samples that reached greater than a 95% detection rate passed preliminary quality control. Additional quality control measures were conducted at UAB to adjust for insufficient data and/or failed QC. 

### 2.4. Epigenetic Age and Age Acceleration

To estimate epigenetic age in offspring, DNAm data were analyzed using the R package ENmix [25]. Beta values were uploaded to the Horvath online calculator in a methylation file after formatting [26,27]. Only CpGs present on the Illumina Infinium HumanMethylation450 BeadChip 450,000 CpG site platform (450K; Illumina Inc., San Diego, CA, USA) that are not on the Infinium MethylationEPIC 850K BeadChip microarray were included in the final file based on the online calculator tutorial [26]. Estimates of DNAm age, age acceleration differences, and age acceleration residuals were output from the online calculator and predicted DNAm age was calculated in R. Age acceleration residuals for the Horvath and Hannum clocks were calculated by regressing the predicted epigenetic age estimates we obtained against chronological age. This method of calculating epigenetic age is based on 353 CpGs that have exhibited correlations with chronological age. Conversely, the Hannum method is based on 71 CpGs [28]. 

### 2.5. Statistical Analysis

Participant demographics are presented as means and standard deviations (SD) for continuous variables and percentages for categorical variables by GDM status. Chi-square or Fisher’s exact tests were performed for categorical variables and *t*-tests for continuous variables. These analyses were performed for mothers and offspring stratified by exposure status. Spearman correlation coefficients were also estimated. Linear regression models were performed to evaluate the association between GDM exposure in mothers and offspring and accelerated epigenetic age. Age acceleration residuals calculated from our linear regression model between predicted epigenetic age and chronological age were utilized as our estimate of EAA. Additionally, we performed multivariable linear regression models to investigate the potential association between epigenetic age residuals and each of the five metabolic biomarkers of interest. These models were adjusted for additional covariates to include chronological age, sex (in offspring only), current BMI, plate, and cell-type estimates. All statistical tests were carried out in R Studio (version 2021.9.2.382) and SAS software (version 9.4 TS1M7 64-bit) with a *P*-value of <0.05.

## 3. Results

### 3.1. Demographics

Participant demographics for mothers and offspring are displayed in Table 1. On average, mothers who had a GDM pregnancy were chronologically older than mothers who did not have a pregnancy complicated by GDM (GDM: 35.7 years; range 24.7–44.9 vs. non-GDM: 32.2 years, range 24.9–47.4 years). This was the opposite for their offspring, where GDM-exposed offspring were younger chronologically (6.53 years; range 4.0–10.6 years) compared to offspring of non-GDM mothers (7.5 years; range 4.2–10.7 years). GDM and non-GDM mothers had similar racial and ethnic distributions (Black: 78.9% non-GDM vs. 89.4% GDM, *P* = 0.0824; Not Hispanic: 97.2% non-GDM vs. 95.5% GDM, *P* = 0.3518). On average, maternal BMI (24.9 kg/m^2^ non-GDM vs. 37.9 kg/m^2^ GDM, *P* < 0.0001) and body fat percent (31.8% non-GDM vs. 42.1% GDM, *P* < 0.0001) for GDM mothers were higher than non-GDM mothers. Mothers with a GDM pregnancy had a slightly higher mean waist-to-hip ratio than non-GDM mothers (0.8 non-GDM vs. 0.9 GDM, *P* < 0.0001). Conversely, offspring of non-GDM mothers had a higher mean WHtZ score than offspring of GDM mothers (0.9 non-GDM vs. 0.4 GDM, *P* = 0.0376). 

For our exposures of interest, mothers with a GDM pregnancy had higher mean HOMA-IR (5.0 ± 4.5; *P* < 0.0001), leptin (73.8 ± 34.3; *P* < 0.0001), fasting glucose (118.23 ± 65.26; p = 0.0008), and fasting insulin scores (17.1 ± 11.8; *P* < 0.0001), whereas mothers with a non-GDM pregnancy had a higher HDL-C score (68.2 ± 13.0; *P* < 0.0001). Children whose mothers had a GDM pregnancy had slightly higher mean leptin (16.8 ± 16.9; *P* = 0.2259) and HDL-C (64.2 ± 10.8; *P* = 0.8316) scores. However, children of non-GDM mothers had a higher mean HOMA-IR score (1.8 ± 2.7; *P* < 0.0001). Scores for fasting glucose (88.2 non-GDM vs. 88.6 GDM, *P* = 7887) and insulin (7.7 non-GDM vs. 7.9 GDM, *P* = 0.9314) were similar for children of non-GDM and GDM mothers. 

Results from the Horvath and Hannum clocks in mothers and offspring were similar. On average, the epigenetic age of mothers for the Horvath clock in the GDM group was higher than in the non-GDM group (non-GDM: 43.4 ± 9.7, GDM: 48.4 ± 8.5; *P* = 0.0018). The mean epigenetic age of mothers for the Hannum clock was higher for GDM mothers compared to non-GDM (non-GDM: 41.3 ± 9.9, GDM: 46.9 ± 8.4; *P* = 0.0005). However, the opposite was seen in the children. Offspring of non-GDM mothers had a higher mean epigenetic age than offspring of GDM mothers for Horvath (non-GDM: 20.2 ± 7.7, GDM: 17.3 ± 8.4; *P* = 0.0605) and Hannum (non-GDM: 20.7 ± 7.4, GDM: 17.6 ± 8.4; *P* = 0.0353) clocks.

### 3.2. Correlations between Epigenetic Age and Chronological Age

To compare chronological age and epigenetic age, we performed correlation analyses for mothers and offspring, presented in Figure 1. Mothers in the non-GDM and GDM groups displayed strong positive correlations between chronological age and DNAm age for the Horvath clock (non-GDM: r = 0.76, *P* = 1.8 × 10^−14^; GDM: r = 0.72, *P* < 2.2 × 10^−16^). The Hannum clock revealed a strong positive correlation for non-GDM and GDM mothers between chronological age and DNAm age (non-GDM: r = 0.79, *P* = 3.1 × 10^−16^; GDM: r = 0.67, *P* = 7.1 × 10^−11^). Offspring of non-GDM mothers had moderate positive correlations, and GDM-exposed offspring had a strong positive correlation between their chronological age and DNAm age for the Horvath clock (non-GDM: r = 0.43, *P* = 0.0011; GDM: r = 0.65, *P* = 1.3 × 10^−8^). For the Hannum clock in offspring, there was a stronger positive correlation for offspring of non-GDM and GDM mothers between their chronological age and DNAm age compared to the Horvath clock (non-GDM: r = 0.53, *P* = 3.2 × 10^−5^; GDM: r = 0.66, *P* = 6.8 × 10^−9^).

### 3.3. Age Acceleration

Residuals for age acceleration were estimated through a general linear regression model with predicted DNAm Age as the outcome and chronological age as the predictor [26,28]. Correlations between the Horvath and Hannum age acceleration residuals and our cardiometabolic risk factors of interest were estimated in Table 2. Only the Hannum age acceleration residuals and HDL-C were found to be significantly correlated in mothers (r = −0.23223, *P* = 0.0063). No other biomarkers from mothers and their offspring were significantly correlated with the Horvath or Hannum age acceleration residual. The Horvath age acceleration residual was negatively correlated with three metabolic biomarkers in mothers and their offspring. Additionally, the Hannum age acceleration residual in mothers was negatively correlated with one metabolic biomarker, while in children, it was negatively correlated with three metabolic biomarkers.

The residuals were used to evaluate associations between exposure to a GDM pregnancy and age acceleration in mothers and offspring. Results from the regression of the association between our metabolic biomarkers of interest with the Horvath and Hannum age acceleration residuals in mothers are presented in Table 3. Horvath age acceleration residual was not associated with any metabolic biomarkers. However, HDL-C was negatively associated with Hannum age acceleration residual (*P* = 0.0244) after adjusting for chronological age, current BMI, plate, and cell type. This association remained after adjustments for BMI were removed (Appendix A). No other cardiometabolic risk factors were associated with the Hannum age acceleration residual in mothers. In the model unadjusted for cell type, no biomarkers were found to be significantly associated with the Horvath or Hannum age acceleration residual (Appendix A). Additionally, we ran a model adjusting for covariates from Table 3 and GDM status and found a marginal association between HDL-C and the Hannum age acceleration residual (Appendix A).

Presented in Table 4 are the results from the regression model examining the association between our metabolic biomarkers of interest with Horvath and Hannum age acceleration residuals in offspring. Of the five metabolic biomarkers evaluated, only fasting insulin was significantly associated with Hannum age acceleration residual (*P* = 0.0488). Removing adjustments for cell type revealed a slight increase in the strength of the association between fasting insulin and Hannum age acceleration residual (*P* = 0.0465). However, this association was removed when adjustments for BMI z-score were not included (Appendix A). The Horvath age acceleration residual was not significantly associated with any metabolic biomarkers. However, leptin was negatively associated with both age acceleration residual estimates even after adjustments for BMI z-score and cell type were removed (Appendix A). Removing adjustments for cell type changed the direction of the association between fasting glucose from positive to negative (Appendix A). A significant association remained between fasting insulin and Hannum age acceleration residual after adjusting for covariates in Table 4 and GDM status (Appendix A). All other metabolic biomarkers in our main and supplementary regression model were positively associated with the Horvath and Hannum age acceleration residual estimates. Additional sensitivity analyses were performed to evaluate these results stratified by sex, however those results were not statistically significant.

## 4. Discussion

In this study, we sought to quantify and compare measures of epigenetic age to chronological age in mothers and their children after a pregnancy complicated by GDM. Further, we sought to evaluate potential associations between two measures of age acceleration from the Horvath and Hannum clocks with metabolic biomarkers. From our analysis of epigenetic age on the Horvath and Hannum clocks, mothers who had a prior GDM pregnancy were older on average compared to non-GDM mothers. However, the epigenetic age of offspring born to GDM mothers was, on average, lower than those born to non-GDM mothers. Our analysis revealed that the Hannum age acceleration residual was associated with HDL-C in mothers independent of BMI but not cell type. In offspring, there was an association between the Hannum age acceleration residual and fasting insulin independent of cell type but not BMI z-score. However, due to our smaller sample size, we were unable to evaluate whether our observed associations remained based on stratification by GDM status. To our knowledge, no other study has assessed the association between age acceleration in mother–child dyads exposed to a GDM pregnancy with metabolic biomarkers. Thus, future research is warranted to better interrogate these underlying mechanisms.

Prior studies of GDM have found notable changes in the maternal and offspring DNAm patterns. Kang et al. found over 300,000 genome-wide differentially methylated CpG sites in maternal blood samples associated with exposure to a current GDM pregnancy [29]. Cord blood samples in this study found similar differential methylation between the exposed and unexposed groups, revealing the potential heritability of DNAm differences associated with exposure to GDM during pregnancy. In comparison, in another study of women sampled mid-pregnancy and one year after giving birth without prior pregnancy complications, maternal epigenetic aging in this cohort was found to have decreased after birth compared to their chronological age [30].

Our study found statistically significant associations between HDL-C and epigenetic age acceleration residuals for the Hannum clock only in mothers. Research has established that mothers with GDM during pregnancy experience significant changes in their metabolic profile during and after giving birth. A prior study found that mothers who experience GDM during pregnancy exhibit lower HDL-C and increasing triglyceride (TG) and total cholesterol (TC) levels throughout their pregnancy compared to non-GDM mothers [31]. In a study by Chodick et al., mothers who experienced a prior GDM pregnancy had lower HDL-C levels and increased TG and TC levels compared to non-GDM mothers, even after adjustments for age, BMI, smoking, and prior lipid levels [32]. HDL-C has been shown to interact with free radicals like lipid peroxides to inhibit LDL oxidation, which has been shown to result in the development of inflammation in the arteries (atherosclerosis) [33]. Low levels of HDL-C have been associated with biological processes that lead to the accrual of oxidized LDL and potential cardiometabolic risk [33]. Prior studies have also examined the anti-oxidant effects of HDL during pregnancy, including those complicated by GDM [34,35,36]. These modifications are marked by increased free radicals and lipid peroxidation, leading to oxidative stress [37]. One study noted that pregnancies complicated by GDM revealed increases in lipid peroxides and a decrease in anti-oxidants [34]. Similarly, another study looking at GDM found that multiple markers of oxidative stress were significantly higher compared to those without GDM [35]. Additional work is needed to gain insight into the role of oxidative stress, inflammation, and accelerated epigenetic aging for pregnancies complicated by gestational diabetes.

Overall, offspring in this study had statistically significant associations between fasting insulin and Hannum epigenetic age acceleration residuals. The literature on the association between exposure to a GDM pregnancy and future cardiometabolic outcomes remains inconclusive in offspring. A study by Tam et al. found that offspring exposed to GDM experienced similar metabolic outcomes as non-GDM-exposed offspring [38]. However, they reported that in utero exposure to increased blood insulin levels resulted in increased offspring BMI and the onset of metabolic syndrome. The HAPO study found GDM-exposed offspring experience increased odds of impaired glucose tolerance compared to non-GDM-exposed offspring, even after adjustments for maternal and child BMI and history of diabetes [39]. Additionally, another HAPO study found that the difference in overweight or obesity in children defined by BMI was not statistically significant [11]. Our findings help contribute to the paucity of research investigating the association between exposure to a GDM pregnancy with future cardiometabolic outcomes and the association between epigenetic age and pregnancy complications like GDM.

Although our study has provided insight into the epigenetic impact of GDM on mothers and offspring, it does have a few limitations. First, our study exposure of interest occurred 4–10 years before DNAm sample collection. Thus, it is possible that the range of time since pregnancy (i.e., 4–10-year range) and the potential contribution of other environmental factors could limit our inference of the association between DNAm age, chronological age, and metabolic biomarkers. However, all samples were assayed at the same time and randomized on methylation assay plates, limiting potential confounding introduced by batch effects. Implementing a cross-sectional design and narrow recruitment timeline lend greater confidence to the estimates of epigenetic age and chronological age for each mother and child in this study. Second, the Hannum clock was initially trained and validated only in adult samples [40]. We cannot exclude the potential for differences that might contribute to the moderate correlations we found in our analysis. However, our study is one of a few that utilized the Hannum clock in a pediatric population, and our estimates of epigenetic age were similar to the Horvath clock. Next, we had a modest sample size for GDM and non-GDM mothers and offspring. Prior studies examining DNAm age in mothers and offspring have had larger sample sizes [16,40]. A smaller sample size may have reduced the power of our analysis of epigenetic age to predict differences between groups. Finally, we observed differences in the direction of the relationship of our metabolic biomarkers of interest with the Horvath and Hannum age acceleration residuals in mothers and their offspring. However, this is consistent with three studies that assessed associations between metabolic biomarkers and a measure of age acceleration [15,16,27].

## 5. Conclusions

In this cohort, mothers with a prior GDM pregnancy exhibited a higher mean epigenetic age than non-GDM mothers. However, this association was not sustained in their offspring. Significant associations were found between the Hannum age acceleration residual and having a lower HDL-C in mothers and higher fasting insulin in their offspring. These associations were altered during our various model adjustments, but marginal associations in mothers and significant associations in offspring remained irrespective of the covariates adjusted. Exposure to a GDM pregnancy poses immediate and future health risks for mothers and their offspring but garnering an understanding of the relationship between GDM and DNAm age may help inform ways to slow epigenetic aging in these populations. Thus, studies evaluating associations between accelerated epigenetic aging and metabolic biomarkers may play a role in determining potential interventions that can help lower the risk for adverse health outcomes experienced after a GDM pregnancy (e.g., T2D and CVD). Future studies on this topic are necessary to examine these findings in diverse and large cohorts, and better elucidate the mechanisms underlying the associations between a GDM pregnancy and cardiometabolic risk factors.

## Figures and Tables

**Figure 1 metabolites-12-01226-f001:**
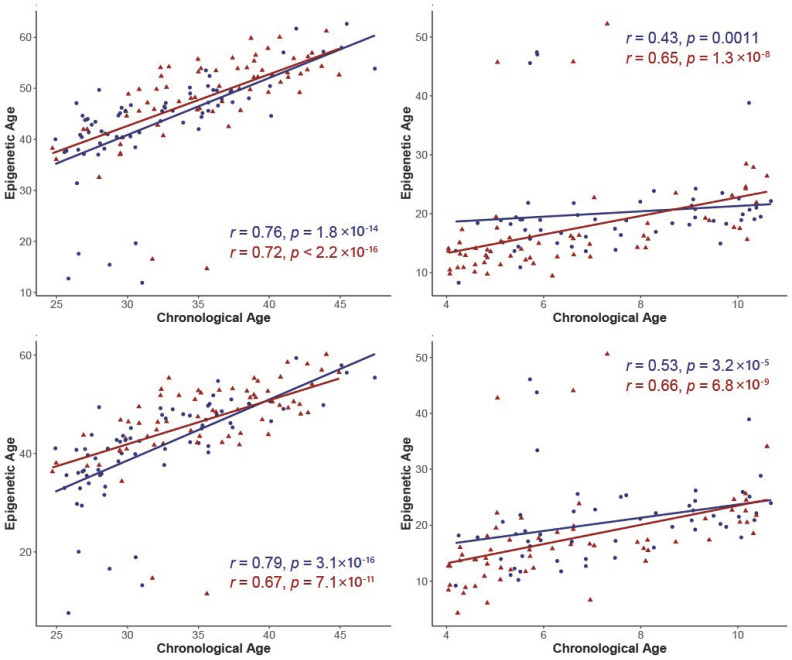
Correlation plots of epigenetic age and chronological age by exposure group: (**A**) Horvath clock in mothers; (**B**) Horvath clock in offspring; (**C**) Hannum clock in mothers; (**D**) Hannum clock in offspring.

**Table 1 metabolites-12-01226-t001:** Participant Demographics for Mothers and Children from the HAPI Study (N = 253).

	Non-GDM		GDM		*P*-Value	
Characteristic	Mothers N = 71	Offspring N = 55	Mothers N = 66	Offspring N = 61	Mothers	Offspring
**Age (years)**					0.0002	0.0161
Mean (SD)	32.2 (5.4)	7.5 (2.0)	35.7 (5.0)	6.5 (2.1)		
**Maternal Race, n (%)**					0.0824	--
Black	56 (78.9)	42 (76.4)	59 (89.4)	54 (88.5)		
White	14 (19.7)	12 (21.8)	6 (9.1)	6 (9.8)		
Asian/AI	1 (1.4)	1 (1.8)	1 (1.5)	1 (1.6)		
**Maternal Ethnicity, n (%)**					0.3518	--
Not Hispanic	69 (97.2)	53 (96.4)	63 (95.5)	58 (95.1)		
Hispanic	1 (1.4)	1 (1.8)	3 (4.6)	3 (4.9)		
Missing	1 (1.4)	1 (1.8)	0 (0)	0 (0)		
**Mom Current BMI (kg/m^2^)**					<0.0001	--
Mean (SD)	24.9 (4.1)	--	37.9 (9.7)	--		
**Body Fat Percent ^a^**					<0.0001	0.3057
Mean (SD)	31.8 (7.3)	23.6 (8.9)	42.1 (6.3)	25.3 (9.6)		
**Waist-to-hip ratio**					<0.0001	0.0245
Mean (SD)	0.8 (0.1)	0.9 (0.1)	0.9 (0.1)	0.9 (0.1)		
**HOMA-IR**					<0.0001	0.9978
Mean (SD)	2.0 (1.6)	1.8 (2.7)	5.0 (4.5)	1.8 (1.9)		
**Leptin**					<0.0001	0.2259
Mean (SD)	38.2 (35.6)	12.9 (17.2)	73.8 (34.3)	16.8 (16.9)		
**HDL-C (mg/dL)**					<0.0001	0.8316
Mean (SD)	68.2 (13.0)	63.8 (11.8)	56.0 (11.9)	64.2 (10.8)		
**Fasting Glucose (mg/dL)**					0.0008	0.7887
Mean (SD)	89.9 (8.4)	88.2 (8.6)	118.3 (65.3)	88.6 (6.9)		
**Fasting Insulin (μU/mL)**					<0.0001	0.9314
Mean (SD)	8.7 (6.2)	7.7 (10.8)	17.1 (11.8)	7.9 (8.0)		
**Child Sex, n (%)**					--	0.9459
Female	--	31 (56.4)	--	34 (55.7)		
Male	--	24 (43.6)	--	27 (44.3)		
**Tanner Stage, n (%)**					--	0.8902
Stage 1	--	46 (83.6)	--	51 (83.6)		
Stage 2	--	7 (12.7)	--	8 (13.1)		
Stage 3	--	1 (1.8)	--	2 (3.3)		
Missing	--	1 (1.8)	--	0 (0)		
**Child BMI z**					--	0.1996
Mean (SD)	--	0.4 (1.1)	--	0.7 (1.3)		
**Child WHtZ**					--	0.0376
Mean (SD)	--	0.9 (1.1)	--	0.4 (1.3)		

^a^ Missing data: Non-GDM Offspring = 1 missing; GDM Offspring = 1 missing. *P*-values for continuous variables were obtained through *t*-tests and categorical variables through chi-square. BMI: Body mass index, HDL-C: High-density Lipoprotein cholesterol, HOMA-IR: Homeostatic model assessment of insulin resistance, GDM: Gestational diabetes mellitus.

**Table 2 metabolites-12-01226-t002:** Correlation of epigenetic age acceleration residuals and metabolic biomarkers in mothers and offspring.

	Mothers (N = 137)				Offspring (N = 116)			
	Horvath Age Acceleration Residual		Hannum Age Acceleration Residual		Horvath Age Acceleration Residual		Hannum Age Acceleration Residual	
	r	*P*	r	*P*	r	*P*	r	*P*
BMI *	0.02713	0.7530	0.08014	0.3519	0.02134	0.8202	−0.00857	0.9273
WHR	0.07456	0.3865	0.04489	0.6025	−0.15946	0.0873	−0.03823	0.6837
Leptin	−0.02402	0.7805	0.05452	0.5269	−0.08996	0.3369	0.01168	0.9009
HOMA-IR	0.06392	0.4580	0.10197	0.2358	0.06193	0.5090	0.06326	0.4999
HDL-C	−0.07286	0.3975	−0.23223	0.0063	−0.04985	0.5951	−0.10348	0.2690
Fasting Glucose	−0.03274	0.7041	0.07880	0.3600	0.14334	0.1248	0.07058	0.4515
Fasting Insulin	0.07759	0.3675	0.11728	0.1723	0.04552	0.6275	0.05466	0.5601

* BMI in mothers measured as kg/m^2^ and in offspring measured as z-score. WHR: Waist-to-hip ratio. HDL-C: High-density lipoprotein cholesterol. HOMA-IR: Homeostatic model assessment of insulin resistance.

**Table 3 metabolites-12-01226-t003:** Multivariable Linear Regression * of the association between metabolic biomarkers with DNAm age acceleration residual in Mothers (N = 137).

	Horvath Age Acceleration		Hannum Age Acceleration	
Variable	β (SE)	*P*-Value	β (SE)	*P*-Value
HDL-C (mg/dL)	−0.073(0.041)	0.0751	−0.088(0.038)	0.0244
Leptin	−0.028(0.022)	0.2038	−0.021(0.020)	0.3155
HOMA-IR	0.090(0.164)	0.5833	0.058(0.154)	0.7056
Fasting Glucose (mg/dL)	−0.005(0.013)	0.7206	−0.008(0.012)	0.4901
Fasting Insulin (μU/mL)	0.053(0.057)	0.3605	0.035(0.054)	0.5145

* Adjusted for chronological age, current BMI, plate, and cell type (CD8T, CD4T, NK, Bcell, Mono, Gran).

**Table 4 metabolites-12-01226-t004:** Multivariable Linear Regression * of the association between metabolic biomarkers with DNAm age acceleration residual in Offspring (N = 116).

	Horvath		Hannum	
Variable	β (SE)	*P*-Value	β (SE)	*P*-Value
HDL-C (mg/dL)	0.030(0.061)	0.6226	0.017(0.051)	0.7488
Leptin	−0.037(0.058)	0.5226	−0.0098(0.049)	0.8420
HOMA-IR	0.461(0.342)	0.1807	0.533(0.286)	0.0648
Fasting Glucose (mg/dL)	0.087(0.088)	0.3259	0.062(0.074)	0.4014
Fasting Insulin (μU/mL)	0.128(0.084)	0.1320	0.143(0.070)	0.0488

* Adjusted for chronological age, sex, BMI z-score, plate, and cell type (CD8T, CD4T, NK, Bcell, Mono, Gran).

## Data Availability

Not applicable.

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
