# Peer review of "Epigenetic Age Acceleration in Mothers and Offspring 4–10 Years after a Pregnancy Complicated by Gestational Diabetes and Obesity"

_metabolites, 2022, doi:10.3390/metabo12121226_

Round 1
Reviewer 1 Report
The authors assessed the association between GDM and epigenetic age acceleration (EAA) in GDM-exposed offspring in. They enrolled 137 mother-child dyads with an index pregnancy 4-10 years before study enrollment. They found that mothers with a GDM and non-GDM pregnancy had strong correlations between chronological age and DNAm age. Offspring of GDM mothers had moderate to strong correlations, whereas offspring of non-GDM mothers had moderate correlations between chronological age and DNAm age. Association analyses revealed a significant association between EAA and fasting insulin in offspring, while HDL-C was the only metabolic marker significantly associated with EAA in mothers. Mothers in the GDM group had a higher predicted epigenetic age and age acceleration than mothers in the non-GDM group. They concluded that the association between EAA with elevated fasting insulin in offspring and elevated HDL-C in mothers suggests possible biomarkers that can better elucidate the effects of exposure to a GDM pregnancy and future cardiometabolic outcomes.
This is a well designed and elegantly presented study reporting important novel data on the effect of GDM on epigenetic age acceleration in GDM-exposed offspring.
Comments:
1. Data on concomitant diseases and medication of mothers could be included.
2. Data on smoking would be important, since tobacco use might be associated with accelerated biological aging including DNAm age.
3. The significant association between HDL-C and EAA is a very interesting finding. Anti-oxidant and anti-inflammatory effects of HDL could be discussed.
Reviewer 2 Report
In this manuscript, authors examining the association of EAA with five metabolic biomarkers of interest and GDM by calculating epigenetic age as a comparator to chronological age in mothers with and without GDM and their offspring. The work is interesting, it has been designed carefully, results are adequately presented.
Question 1
The word race is not adequate, delete this word from the results lines 160-161, and from the table; you could leave only the comparison between ethnic groups.
Question 2
In your previous work” Mother‐child cardiometabolic health 4–10 years after pregnancy complicated by obesity with and without gestational diabetes”, you considered three groups or OB OB-GDM and NW. Explain why in this work you only analyzed OB-GDM and NW.
Question 3
To evaluate the results on the five metabolic biomarkers in offspring segregated by sex.
